# Predicting metal-protein interactions using cofolding methods: Status quo

## Abstract

Metals play important roles for enzyme function and many therapeutically relevant proteins. Despite the fact that the first drugs developed via computer aided drug design were metalloprotein inhibitors, many computational pipelines still discard metalloproteins due to the difficulties of modelling them computationally. New "cofolding" methods such as AlphaFold3 (AF3)[1] and RoseTTAfold-AllAtom (RFAA)[2] promise to improve this issue by being able to dock small molecules in presence of multiple complex cofactors including metals or covalent modifications. Here, we analyze the current status for metal ion prediction using these methods. We find that currently only AF3 provides realistic predictions for metal ions, RFAA in contrast does perform worse than more specialized models such as AllMetal3D in predicting the location of metal ions accurately. We find that AF3 predictions are consistent with expected physico-chemical trends/intuition whereas RFAA often also predicts unrealistic metal ion locations.

## 1. Introduction

Metals are versatile and indispensable cofactors for many proteins and a lot of DNA/RNA chemistry[3]. A major category of biological metals are transition metals such as zinc (used in enzymes or for structural stability such as e.g in zinc finger domains or as Lewis acid in catalysis), as well as iron and copper (e.g for electron transport). Earth alkali ions such as magnesium (used in ATP and nucleic acid chemistry) and calcium (used for signal transduction or coagulation) also play important roles. About 10% of enzyme reactions depend on zinc alone [4]. For this reason many metalloproteins are also therapeutically relevant targets. Among the first drugs developed using structure based drug design were

Captopril (1981)[5] and dorzolamide (1995) [6] that both target zinc enzymes. Yet even today it is computationally still difficult to treat transition metal ions using classical modelling techniques due to their complex electronic structure. Only quantum mechanics based methods can faithfully model all effects relevant for a proper description of (transition)metal ion coordination geometry[7]. Classical force fields such as Amber and CHARMM or knowledge-augmented force fields such as the Autodock Vina scoring function or the Rosetta energy function are inadequate to model most metals found in biology[8]. Luckily, there is ample experimental data available on metal ions in the protein data bank (PDB) therefore it was quickly noted that modern structure prediction models such as AlphaFold2[9] often predict the holo (i.e the binding) form of metal ion binding sites since those usually also have a strong coevolution signal[9]. This has given rise to tools such as AlphaFill[10], that transplant metal ions from experimental structures with high homology to predicted models.

Deep learning based tools can now also predict metal ion location from single structures[11] and are sensitive even to small side chain rearrangments[12].

Here, we investigate how RoseTTAfold-All Atom (RFAA)[2] and AlphaFold3(AF3)[1] as new cofolding methods handle structure prediction in presence of metal ions in comparison with state of the art specialized models for metal ions operating on a given structure.

### 1.1. Background

#### 1.1.1. ROSETTAFOLD-ALL ATOM

RoseTTAfold-AllAtom (RFAA) is an extension of the architecture of RoseTTAfold2[13] extending the number of tokens for the 1D track, adding bond distances to the positional encoding of the 2D track and adding chirality inputs to the structure module of the 3D track.

Metal ions are provided to RFAA as a as a single atom ligand similar to all other small molecules. Since metal ions only consists of a single atom, they do not have their own canonical frame in RFAA. Therefore, the network does not receive a frame input and no loss is calculated with respect to the frame of the ion. The error of the placement of the ion with respect to the other frames in the structure is still

[1]Anonymous Institution, Anonymous City, Anonymous Region, Anonymous Country. Correspondence to: Anonymous Author <anon.email@domain.com>.

Preliminary work. Under review by the Machine Learning for Life and Material Sciences Workshop at ICML 2024. Do not distribute.

included in the loss.

Training of RFAA is split into three steps and the dataset of protein-metal complexes is not sampled in the first training step. In the second step, 10% of training examples are protein-metal complexes, while in the final fine-tuning step only 3% of examples are protein-metal complexes. RFAA samples all important biological metal ions but it is not reported what the sampling statistics for the respective metal categories are and the sampling frequency might deviate from the statistics in the PDB.

### 1.1.2. ALPHAFOLD3

AlphaFold3[1] is an improvement over AlphaFold2 [9] and is able to handle organic ligands, nucleic acids and metal ions. The model simplifies the original architecture by replacing the Evoformer with the Pairformer module and the invariant point attention with a simpler diffusion based architecture. No backbone frame loss is used anymore. The code is not available but predictions can be run using a webserver for select ligands and ions.

## 2. Methods

Predictions were run using the two test sets from Metal3D [11] for zinc and pdb codes MN, NI, CO, CU, FE2, FE, ZN, NA, K, CA, MG. This dataset (887 proteins) might include structures that RFAA and AF3 were trained on due to the training cutoff. For each PDB identifier in the dataset, the sequence of the chains of the first biological assembly was used. As small molecule inputs, the same number of ions as for the investigated metal were added. No other small molecule or DNA/RNA was modelled.

The official RFAA implementation (8.March 2024) was used to run predictions on a local workstation with a GTX 3090 Ti with 24GB of memory. A bugfix for concatenating multiple sequence alignments (MSAs) was applied. Predictions were run using default settings for RFAA. For 248 proteins no results could be obtained because of memory limitations.

We used the deep-learning based Metal3D [11] and a newer retrained version called AllMetal3D, which used input from the above mentioned metals to predict a general metal binding probability for any of considered metals in a single output channel. Except for the training data, AllMetal3D uses the same architecture as Metal3D[11] and was trained in the same way.

For AlphaFold3 predictions were run using the public webserver. 5 proteins containing the possible metals (all except NI, FE2) could not be run on the server (e.g due to containing X in the sequence). For all analysis out of the 5 predicted structures for a given seed the one with the best `ranking_score` is used. All predictions were run with

seed `1593933729`.

Precision and recall was computed as detailed in [11].

## 3. Results

### 3.1. Performance on zinc

We first investigated how well RFAA and AF3 perform for $Zn^{2+}$, which is the most abundant transition metal in biology.

On the Metal3D zinc set RFAA performs worse than the more specialized models Metal3D and AllMetal3D (Figure 1). AF3 has similar performance compared to the specialized models with high confidence especially on physiological sites with 3 or more unique residues coordinating the metal ion. AF3 makes most predictions with high confidence (Figure A1). If only highly confident zinc predictions are taken (pLDDT >75) RFAA does only predict 9 out of 189 zinc sites. Overall RFAA can only find 38% of the zinc sites that have at least 3 unique coordinating protein residues. Precision and recall are lower when evaluating on all zinc sites with fewer unique coordinating residues.

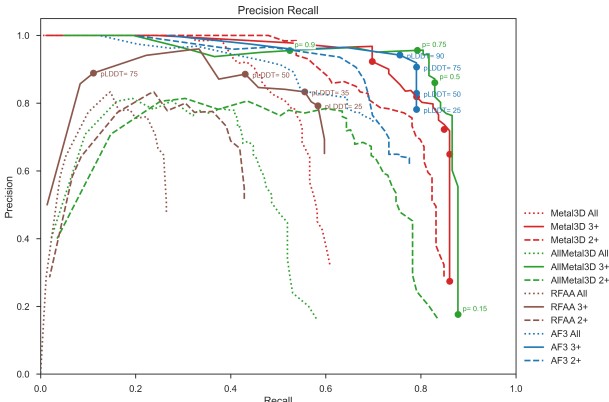

*Figure 1.* Precision versus Recall computed using the Metal3D zinc test for RFAA, AF3, Metal3D and AllMetal3D for all experimentally resolved $Zn^{2+}$ or coordinated by at least 2+ respectively 3+ unique protein residues.

### 3.2. Consistency with physico-chemical intuition

We also wanted to test whether cofolding methods make predictions on metal ion location consistent with expected physical trends. For this we tested mutations to the active site of human carbonic anhydrase (hCA)[14], which is a well studied metalloenzyme.

We made mutations to the input sequence changing the first and second shell metal binding residues to alanine individually or together. Since the retrieved MSA still contains unaltered homologues for RFAA we also ran the H{94,96,119}A,E106A,T{199,198}A,Y7A variant editing

all rows of the MSA of the wildtype(WT) to generate an all-alanine MSA similar to Stein et al.[15] (Table A1). For AF3 no options exist to modify the input MSA.

For the WT sequence RFAA can predict the location of zinc with high positional accuracy and high pLDDT albeit lower than specialized models (1.2Å compared to AllMetal3D 0.28Å). AF3 has very high confidence and low deviation from the experimental position with 0.1Å.

For RFAA generating the input MSA with all first shell residues mutated to alanine increases the RMSD of the predicted coordinates with respect to the WT from 0.5 to 0.8Å. The location of the zinc binding site is maintained with 2.4 Å deviation with a big decrease in predicted confidence (pLDDT 28). In this H{94,96,199}A mutant the zinc binding site is formed by T199, T198 and E106. When those are mutated to alanine as well before running the MSA generation the position of the zinc is still in the central pocket of hCA with 4Å distance deviation to the experimental location. Now the zinc is coordinated by Y7. Using the input MSA with all rows mutated and the Y7A mutation, RFAA performs better to predict the global structure of the protein than running MSA generation with a mutated sequence, yet the positioning of the metal inside the central pocket of the protein is maintained (Table A1). The predicted location of the zinc ion is not confident with pLDDT <20 in both cases. For the edited MSA, RFAA predicts the zinc as being coordinated by T199A, W16 and W5, with two tryptophan residues, i.e. residues with very low zinc coordination propensity. Moreover, their side chains interact with zinc not via the indole nitrogen in the side chain but with the aromatic hydrogens in the side chain. AF3 has on average lower RMSD than RFAA for the whole protein and also lower deviation compared to the experimental position of the metal. For single alanine mutations distance deviation noticeably increases. When all residues are mutated to alanine AF3 picks a reasonable alternative binding site (H4, H64) that AllMetal3D does not detect due to the unfavorable rotamer conformation of these residues in the native crystal structure.

## 3.3. Stoichiometry prediction

A disadvantage of current cofolding methods is that the binding stoichiometry needs to be specified beforehand. Sequence based metal ion predictors exist (e.g based on ESM embeddings[16]) but *a priori* it is difficult to determine how many and which metal ions should be predicted. Especially for transition metals most metal binding sites have little selectivity and are often also crystallized with different metals than the biologically relevant ones (e.g to crystallize an enzyme-substrate complex with a metal that does not catalyze the reaction). We therefore also investigated the case where more metals than a physiological amount are used

as input. A physically sound model should occupy each binding site with one metal only and with high confidence and place excess metals in alternative sites that might be populated at high metal concentration.

### 3.3.1. EXCESS METAL IONS:

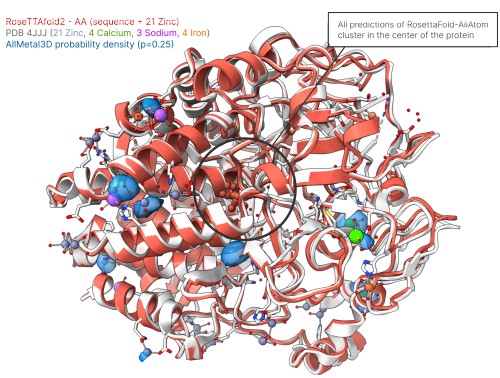

*Figure 2.* Extreme case of a metal saturated structure in the PDB. 4JJJ contains 21 $Zn^{2+}$, 4 $Ca^{2+}$, 3 $Na^+$ and 4 $Fe^{3+}$ ions. AllMetal3D probability density (blue) at probability isovalue 0.25. AllMetal3D does identify all but 9 unique binding sites with one or more metals. RFAA places all 21 zincs inside the protein (black circle)

In the testset of Metal3D, there is a structure of *Thermobifida fusca* Cel48A[17] (PDB entry 4JJJ) refined to contain 21 $Zn^{2+}$, 4 $Ca^{2+}$, 3 $Na^+$ and 4 $Fe^{3+}$ ions. We let RFAA and AF3 predict the structure of the sequence of the biological assembly with the same number of zincs ions as in the experimental structure not including the other metal ions. In contrast to AllMetal3D, AF3 and Metal3D, RFAA cannot accurately place any of the predicted zinc ions at one of the many experimentally resolved metal sites (Figure 2). In fact, the closest distance to any of the predicted zinc locations is 16.4Å. However, overall the quality of the protein structure prediction by RFAA is high ($C_\alpha$ RMSD prediction - crystal 0.57 Å). AF3 predicts the structure of Cel48A well and does identify all but 7 of the experimental $Zn^{2+}$ binding sites similar to AllMetal3D that identifies all but 9 sites using p>0.25.

### 3.3.2. ALTERNATIVE SITES:

In the case of hCA, we ran a prediction for the WT sequence with 5 zinc ions and analyzed the predicted positions for both RFAA and AF3. The quality of the prediction of the protein structure itself remains high for both RFAA($C_\alpha$ RMSD of 0.48 Å) and AF3 ($C_\alpha$ RMSD of 0.18 Å). RFAA predicts all metal ions in the active site close to the experimental location of zinc with distance deviations of 0.7, 1.9, 2.1, 2.9 and 3.3 Å to the (single) experimental location. pLDDT is however low for all predictions (31.2 ± 2.2). For a single $Zn^{2+}$ RFAA has much higher confidence (Table

A1). When predicting the hCA-1ZN complex using AF3 in all 5 predicted models the metal ion is in the exact same position (0.1Å, pLDDT 98.98 ±0.00). When predicting the hCA-5ZN complex, in all 5 AF3 models, the first $Zn^{2+}$ maintains its position 0.1 ±0 Å of the experimental position with pLDDT 98.96 ±0.00. The other 4 $Zn^{2+}$ are predicted by AF3 in realistic positions with suitable coordination partners (e.g H36, D34 & H4, H64 & H4, D19) or single surface residues such as D175 or D190 with some site divergence between the models. The predicted confidence is consistent with the quality in binding motif and number of coordinating residues and is directly reflected in the pLDDT. For the sites with multiple coordinating residues pLDDTs are 78.80 ± 0.82, 69.19 ± 2.44 or 70.83 ± 3.00 across all 5 predicted models. For the predicted sites coordinated by just one residue pLDDT is 38.45 ± 6.58.

### 3.4. General performance for biologically important metals

Specialized metal ion predictors such as Metal3D have shown that predictors trained on $Zn^{2+}$ also can transfer to other rarer but chemically similar transition metal ions such as $Fe^{2+}$ or $Cu^{2+}$[11]. We used the set of structures which were used to evaluate the selectivity of Metal3D[11] and predicted them using RFAA and AF3 using the same number of metals as present in the biological assembly.

We first investigated the mean pLDDT of the predicted ions and sequences. Overall, since the mean pLDDT for the protein structures themselves is high (RFAA > 80, AF3 > 90), we chose to align the predicted structures using CEAlign on the experimental structures for the analysis. We also analyzed the pLDDT of the residues in the vicinity of the predicted metal ion within 5Å (RFAA 83±9.5, AF3 93.12±0.07) compared to the mean protein pLDDT (RFAA 80±6.2). This difference is statistically significant (RFAA $p$ =4.2E-7, AF3 $p$ =1.9E-16). For RFAA metal ion predictions, we observe two distinct peaks of pLDDT for most ions with the majority of ions being predicted with low pLDDT of around 25 (Figure A2). A second peak exists for most ions around pLDDT 70. AF3 predictions are on average very confident with only few low confidence predictions made (Figure A1). Distributions are plotted in groups according to chemical similarity except for manganese which nominally is a transition metal but is also known to bind in similar fashion to magnesium.

Analyzing individual performance in predicting metal sites defined as predictions within 5 Å of the experimental location also shows that RFAA is inferior compared to AF3 and AllMetal3D with similar trends with respect to precision and recall on transition metals, earth-alkali and alkali metals (Figure 3). For $Zn^{2+}$ and $Ni^{2+}$ RFAA performs worse compared to other transition metals which is surpris-

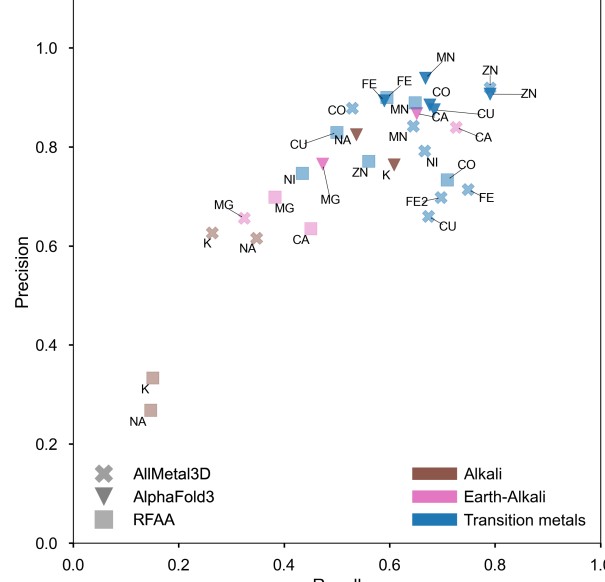

*Figure 3.* Precision versus recall for different metal ions computed with RFAA (pLDDT>0.25), AlphaFold3(pLDDT>0.75) and All-Metal3D(p=0.75) for all metals coordinated by at least 3 unique protein residues with full occupancy

ing given that $Zn^{2+}$ is the most abundant transition metal in the PDB. Another notable difference can be observed for $Ca^{2+}$ where AllMetal3D and AF3 perform much better than RFAA. A difference exists also for $Mg^{2+}$, $K^+$ and $Na^+$ where AllMetal3D is inferior to AF3.

### 3.5. Hard cases

Two difficult cases where dataset bias can play a role where investigated using AF3 only.

3.5.1. CONFORMATIONAL CHANGES IN ZINC PROTEINS:

Conformational changes in proteins can be triggered by metal binding (e.g metal mediated assembly) or even by metal replacement. An example is the Ros87 protein [18], a prokaryotic zinc finger. Ros87 is an interesting test case because it does form its own singular cluster at 30% sequence identity within the PDB and the only structure is an NMR structure without zinc modelled yet with the residues preorganized for metal binding. The structure is likely included in the training data for RFAA and AF3. We used AF3 to predict the Ros87 structure in absence of metals, and in presence of $Zn^{2+}$ or $Zn^{2+}$ and $Cu^{2+}$ or just $Cu^{2+}$. Experimentally it is observed that the protein is unfolded in absence of $Zn^{2+}$ or in presence of $Cu^{2+}$ (likely because of the redox-induced formation of disulfide bridges)[18].

AF3 correctly predicts the structure of Zn-Ros87 and also incorrectly predicts the structure of apo-Ros87 as folded

indicating a preference for folded proteins as they occur in the PDB. AF3 also predicts an extra $\alpha$-Helix for residues 64-75 that are clearly unstructured in the NMR conformational ensemble (Figure A3). The Cu-Ros87 and Cu-Zn-Ros87 also are predicted to adopt the same folded structure with copper being placed in the metal site instead of zinc.

### 3.5.2. Testing metal selectivity

Choi and Tezcan [19] developed a protein assembly that selectively binds $Co^{2+}$ over $Cu^{2+}$ even if normally $Cu^{2+}$ binds more tightly to proteins according to the Irving Williams series [19]. The structure was released after the training cutoff for AF3. We predicted the $Co^{2+}$-HEC, $(Co^{2+})_2$-$(Cu^{2+})_2$-HEC, $(Cu^{2+})_2$-HEC complexes using AF3. AF3 gets the individual domains right but the relative orientation of the two domains interacting via the metal ions are not correctly predicted (Figure A4).

## 4. Discussion

Our results show that RoseTTAFold2-All Atom is an inferior metal ion predictor compared to more specialized tools such as AllMetal3D or AlphaFold3. RFAA can predict physiological sites for transition metals with reasonable accuracy but struggles for other biologically important metals such as $Ca^{2+}$ or $Mg^{2+}$. Metals are often predicted with low pLDDT which prevents the use of the confidence metric to interpret the results. AF3 in contrast mostly offers great performance on the combined problem of prediction of the structure and metal location on par with specialized predictors just trained on metal ion location prediction.

From our testing with mutated inputs it seems that RFAA always places metals in pockets (even if no suitable binding residues are in the pocket). It is also notable, that if more than the stoichiometric amount of metal ions is added, the model in some cases predicts metals with low confidence clustered at random positions overlapping with the protein. This points to a lack of an internal representation for physical interactions with metal ions, which could be used to place the ions. It is well known that structure prediction models such as AlphaFold2 can take into account energetic frustration which can be exploited for predicting ligand binding sites [20]. The reason for the performance difference of RFAA versus AF3 for metal ions is likely the non-inclusion of a direct frame loss function for the metals in RFAA and the relatively low sampling of protein-metal complexes during training. Abramson et al. [1] do not report specifically oversampling protein-metal complexes for AF3 so it is likely enough to just sample them at the frequency reported in the PDB.

The tests we conducted with mutations to the metal binding residues also show that structure prediction software can be sensitive to both global and local sequence context. If one removes the residues coordinating the metal ion directly, the model still confidently predicts the global structure but chooses a different more likely local context for the metal ion. While AlphaFold2 was to a certain degree insensitive to point mutations, it is encouraging to see that AlphaFold3 is sensitive to few point mutations for metal ion location prediction.

However, for difficult cases AlphaFold3 still has a bias from the structures it has seen during training where the core of conformationally flexible proteins such as Ros87 which is unstructured in absence of $Zn^{2+}$ is predicted as folded with high confidence even when predicted in absence of $Zn^{2+}$. Ros87 is a good example of a simple transcription factor that is folded in absence of DNA, but needs zinc to be stable. [18, 21, 22] Currently, it is impossible to disable use of templates in the AF3 server, so it cannot be ruled out that the template influences the prediction steering it towards the folded state.

Metal ion binding motifs also have low variability in general (e.g for $Zn^{2+}$ just 62 different first shell binding combinations exist in the PDB with more than 50 examples [11]) and AlphaFold3 does not seem to be able to generalize very well to new motifs such as the one in [19] predicting the interaction of two protein domains via a new kind of metal ion binding site in wrong geometry.

A further limitation is that the stoichiometry of metal ions needs to be specified beforehand for both AF3 and RFAA. A possible strategy to improve both RFAA and AF3 could be to not add metals a priori and have possible metal ions predicted in one of the later recycles. For RFAA, metal ions could additionally be treated as a special form of covalent modification in atomized residues such that a frame (and therefore also a loss) can be computed for them directly when training. This is similar to the strategy used in Rosetta where for example zinc is tethered to a histidine during sequence design[23].

## 5. Conclusion

AlphaFold3 demonstrates that methods to predict the structure of a protein together with its metal ion ligands now are at the level of more specialized predictors such as AllMetal3D if the stoichiometry of binding is known. RoseTTAfold All-Atom does not perform at the same level likely due some design choices when defining the loss functions used to train the model. The sensitivity of AlphaFold3 to few point mutations in the metal-coordinating residues is encouraging. At the same time limitations due to the PDB containing mainly ordered proteins are also evident even for relatively simple proteins where the metal ion mediates the conformational flexibility or oligomeric state of proteins.

## 6. Data Availability

All predictions and scripts to generate figures will be made available on Zenodo. RFAA predictions will be available under CC BY, Code under MIT. AF3 predictions will be available under AlphaFold Server Terms.

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

# A. Appendix

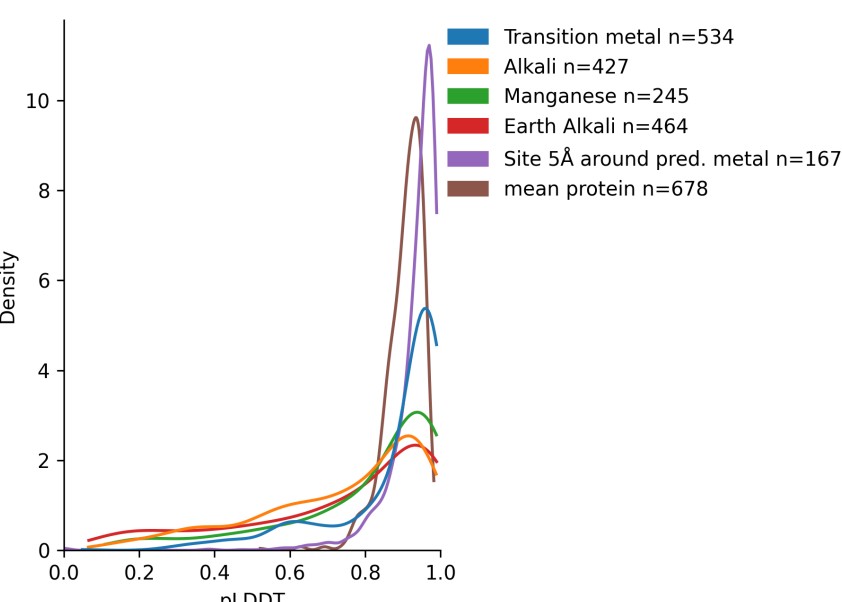

*Figure A1.* pLDDT distribution for metal classes in selectivity set predicted by AF3. pLDDT is also computed as average over all residues in the protein or all residues with any atom within 5 Å of the predicted metal location

*Table A1.* Influence of mutations on predicted metal ion location in hCA. Mutations are the ones made to the input sequence before MSA generation. pLDDT of the predicted $Zn^{2+}$ and euclidean distance to experimental location. $C_\alpha$ RMSD of protein computed after alignment to PDB 2CBA using CEAlign.

| MUTATION | METHOD | PLDDT | DIST (Å) | RMSD PROTEIN(Å) |
|---|---|---|---|---|
| WT | RFAA | 78 | 1.2 | 0.49 |
| H94A | RFAA | 52 | 2.0 | 0.95 |
| H96A | RFAA | 75 | 1.4 | 0.48 |
| H96C | RFAA | 80 | 1.4 | 0.47 |
| H119A | RFAA | 77 | 1.3 | 0.47 |
| H119Q | RFAA | 77 | 1.5 | 0.55 |
| T199A | RFAA | 78 | 1.3 | 0.44 |
| H{94,96,119}A | RFAA | 28 | 2.4 | 0.84 |
| H{94,96,119}A, E106A, T199A | RFAA | 24 | 3.5 | 0.53 |
| H{94,96,119}A, E106A, T{199,198}A | RFAA | 23 | 4.0 | 0.56 |
| H{94,96,119}A, E106A, T{199,198}A, Y7A | RFAA | 15 | 6.3 | 1.44 |
| H{94,96,119}A, E106A, T{199,198}A, Y7A (EDITED MSA) | RFAA | 20 | 6.2 | 0.53 |
| WT | AF3 | 98.98 | 0.1 | 0.18 |
| H94A | AF3 | 89.33 | 0.1 | 0.26 |
| H96A | AF3 | 97.88 | 0.7 | 0.19 |
| H96C | AF3 | 97.65 | 0.6 | 0.17 |
| H119A | AF3 | 98.34 | 0.5 | 0.18 |
| H{94,96,119}A | AF3 | 72.46 | 13.4 | 0.20 |
| H{94,96,119}A, E106A, T199A | AF3 | 75.06 | 13.1 | 0.22 |

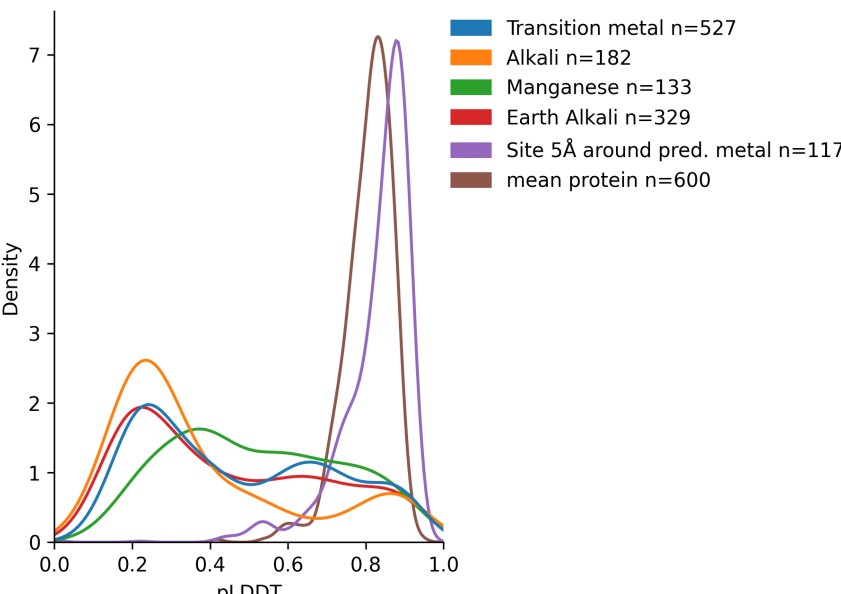

*Figure A2.* pLDDT distribution for metal classes in selectivity set predicted by RFAA. pLDDT is also computed as average over all residues in the protein or all residues with any atom within 5Å of the predicted metal location

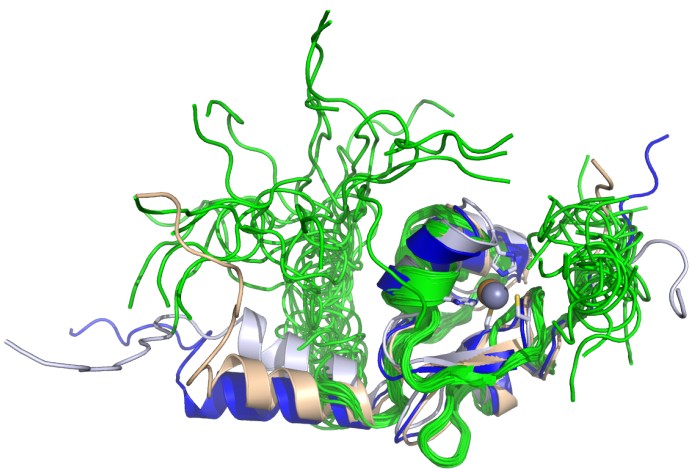

*Figure A3.* Ros87: 2JSP NMR Ensemble(green) of Ros87 protein($Zn^{2+}$ not modeled), light orange: AF3-$Cu^{2+}$, lightblue: AF3-$Zn^{2+}$, darkblue: AF3-apo prediction

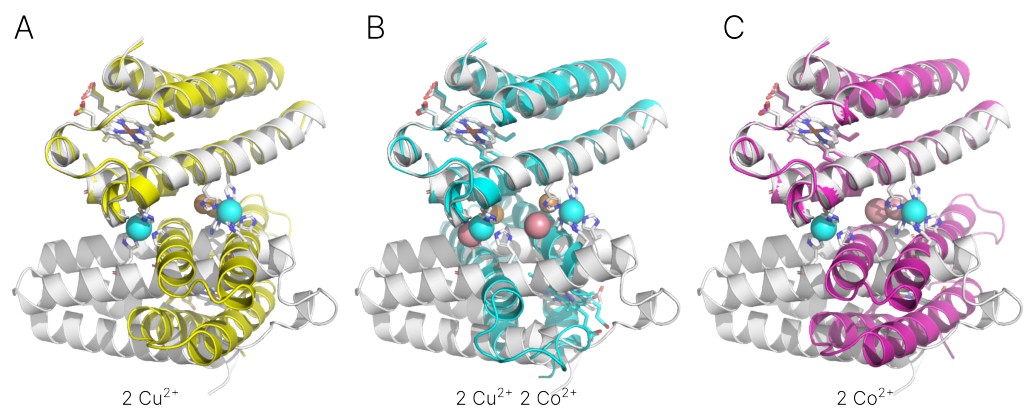

*Figure A4.* Anti-Irving Williams selective metal binding protein by Choi and Tezcan [19]: **A** AF3 prediction with 2 HEC and 2 $Cu^{2+}$ **B** AF3 prediction with 2 HEC, 2 $Co^{2+}$, 2 $Cu^{2+}$ **C** AF3 prediction with 2 HEC and 2 $Co^{2+}$. Native $Co^{2+}$ in cyan, predicted $Co^{2+}$ in lightred, predicted $Cu^{2+}$ in orange. Experimental structure 7MK4 in white. Heme C in sticks.