# OpenReview forum: "Predicting metal-protein interactions using cofolding methods: Status quo"
_ICML.cc/2024/Workshop/ML4LMS — ML4LMS Poster_

### Official Review · Reviewer_TQpW · 2024-06-02
**Interesting investigation in an understudied area**

**Rating:** 6
**Confidence:** 3

**Review:**

In this paper, the authors investigate the prediction and placement of metal ions by structure prediction methods, namely AlphaFold3 and RoseTTAFold-AA. Unsurprisingly, Metal3D (a tool specialized for this task) does the best in a wide range of tasks. The paper provides interesting perspectives on several areas that merit a marginal acceptance, but there are several aspects of the paper that can be improved.

1. There seems to be a metal-dependent effect on prediction performance than strictly a methods-dependent one (Fig 3). For example, ZN, MN and MG tend to have similar metrics across the board, while some metals like FE clearly have lower precision for AllMetal3D. Have the authors checked the relative availability / diversity of these metal complex structures in the PDB? e.g. are there just more ZN structures than FE, and that's why precision is higher?

2. The authors make a really good point in the discussion about proteins like Ros87 (e.g. tools like AF folding it into place without the metal ion, when it shouldn't happen in theory). Can the authors discuss some more examples please as this would help the community?

3. As denoted in section 3.5.1, if there are some structures from the Metal3D set suspected to be in the training sets of RFAA and AF3, can the authors procure a small, 'recent structures' benchmark for a more blind, "out of distribution" type of validation?

4. I'm not quite sure I understand what Figure 2 is supposed to convey. It could maybe do with splitting the true and model structures and doing a more zoomed-in case-by-case of each metal binding site and how excess metal ion conditions affect predictions.

5. The paper's writing is quite haphazard – it has some grammatical and typographic errors and could brush up for easier readability. For example,

* "Two difficult cases where dataset bias can play a role where [were?] investigated using AF3 only."
* "Predictions were run using the two test sets from Metal3D [11] for zinc and pdb codes MN, NI, CO, CU, FE2, FE, ZN, NA, K, CA, MG" <- why is "zinc" and "ZN" repeated, and in some parts of the manuscript we have Zn^{2+} (in superscript). Could be helped with consistency
* "For all analysis out of the 5 predicted structures for a given seed the one with the best ranking_score is used." Maybe this could be re-written to something like "For all analyses, we predict five structures with one seed. We then select the decoy with the best ranking_score metric". I realise it's a small adjustment, but breaking sentences down can be helpful; especially if the sentences are run-on!

---

### Official Review · Reviewer_gRsN · 2024-06-11
**evaluation paper with well designed tests and informed metrics**

**Rating:** 8
**Confidence:** 3

**Review:**

While the paper does not propose a novel methodology, it provides a good evaluation of how AF3 and RFAA handle structure prediction in the presence of metal ions compared to specialised methods developed for this very complicated task that could account for the complex electronic structure of transition metal ions. This gives an overview of the capabilities of SOTA co-folding methods and prospects for their wider adoption in the industry.

The authors demonstrate deep knowledge of the application domain, presenting clear metrics and modification strategies to test the performance of the models. They also present different test suites, such as consistency of predictions for metal ion location with expected physical trends, binding stoichiometry, etc., and look into hard cases like conformational changes in Zinc proteins and testing metal selectivity.

The discussion section shows interesting hypotheses behind the differences across the performance of the methods. Particularly in the following sentence, “The reason for the performance difference of RFAA versus AF3 for metal ions is likely the non-inclusion of a direct frame loss function for the metals in RFAA and the relatively low sampling of protein-metal complexes during training”, it would be good to address why AF3 is performing better for metal ions despite getting rid of the frames altogether.

I consider it very important for the ML community to embrace high-quality metrics and testing of the ML methods designed for specific domains, not just generic benchmarks. When integrating these methods into real applications, good metrics are key; they can guide model improvements, cast light on model generalizability, and assess the value they can bring.

---

### Official Review · Reviewer_yo4Y · 2024-06-12
**Review - Predicting metal-protein interactions using cofolding methods: Status quo**

**Rating:** 7
**Confidence:** 3

**Review:**

The paper investigates the performance of cofolding methods, specifically AlphaFold3 (AF3) and RoseTTAfold-AllAtom (RFAA), in predicting metal-protein interactions. It benchmarks these methods against specialized models such as Metal3D and AllMetal3D, focusing on the accuracy of metal ion placement, consistency with physico-chemical intuition, and the ability to predict metal stoichiometry.

Strengths: This paper has 4 main strengths, (1) Relevance and Novelty, (2) Comprehensive Benchmarking, (3) Methodological Rigor and (4) Insightful Analysis.

The topic is timely and addresses a significant gap in computational biology regarding the prediction of metal-protein interactions. The focus on cofolding methods, which are relatively new, adds novelty. The study provides a thorough comparison of state-of-the-art models (AF3, RFAA) with specialized tools (Metal3D, AllMetal3D) across various metrics, including precision, recall, and consistency with physico-chemical principles. The experimental setup is detailed and methodologically sound, with well-defined test sets and appropriate metrics for evaluation. The paper offers valuable insights into the limitations of current cofolding methods, particularly RFAA, in accurately predicting metal ion locations and handling metal stoichiometry.

Weaknesses: This paper has 3 main weaknesses, (1) Lack of Implementation Details for AF3, (2) Memory Limitations and (3) Incomplete Data Reporting.

Since AF3 is not open-source, the study relies on a webserver for predictions. This limits reproducibility and the ability to extend the analysis to other researchers. RFAA's performance is hindered by memory constraints, which resulted in the exclusion of a significant portion of the test set (248 proteins). This may bias the results and limits the generalizability of the findings. The paper mentions that the training statistics for different metal categories in RFAA are not reported, which could affect the interpretation of the results.

Suggestions for Improvement:

1. Clarify Training Data Overlap: Provide a more detailed analysis of potential overlaps between training and test sets, and discuss how this might impact the results.

2. Report Metal Category Statistics: Include detailed statistics on the sampling of different metal categories during RFAA training to enhance transparency.

3. Address Memory Constraints: Discuss potential solutions or workarounds for the memory limitations encountered with RFAA to provide a more balanced comparison.

4. Expand on AF3 Implementation: Although the code is not available, provide more insights into the architecture and functioning of the Pairformer module in AF3.

My recommendation:

This paper presents a valuable contribution to the field of computational biology and structural bioinformatics. The benchmarking of cofolding methods for metal-protein interaction prediction is thorough and provides important insights. Minor revisions to address the reporting of metal category statistics and potential biases due to memory limitations in RFAA would strengthen the manuscript.